# Is child anemia associated with early childhood development? A cross-sectional analysis of nine Demographic and Health Surveys

**Rukundo K. Benedict** [1]*, **Thomas W. Pullum** [1], **Sara Riese**[1], **Erin Milner**[2]

**1** The Demographic and Health Surveys Program, ICF, Rockville, MD, United States of America, **2** Public Health Institute/ USAID Sustaining Technical and Analytical Resources, Washington, DC, United States of America

* rukundo.benedict@icf.com

**Data Availability Statement:** All Demographic and Health Surveys datasets are publicly available from https://dhsprogram.com/data/available-datasets.cfm.

## Abstract

Anemia is a significant public health problem in many low- and middle-income countries (LMICs), with young children being especially vulnerable. Iron deficiency is a leading cause of anemia and prior studies have shown associations between low iron status/iron deficiency anemia and poor child development outcomes. In LMICs, 43% of children under the age of five years are at risk of not meeting their developmental potential. However, few studies have examined associations between anemia status and early childhood development (ECD) in large population-based surveys. We examined the associations between severe or moderate anemia and ECD domains (literacy-numeracy, physical, social-emotional, and learning) and an overall ECD index among children age 36–59 months. Nine Demographic and Health Surveys (DHS) from phase VII of The DHS Program (DHS-7) that included the ECD module and hemoglobin testing in children under age five years were used. Bivariate and multivariate logistic regressions were run for each of the five outcomes. Multivariate models controlled for early learning/interaction variables, child, maternal, and paternal characteristics, and socio-economic and household characteristics. Results showed almost no significant associations between anemia and ECD domains or the overall ECD index except for social-emotional development in Benin (*AOR* = 1.00 *p* < 0.05) and physical development in Maldives (*AORs* = 0.97 *p* < 0.05). Attendance at an early childhood education program was also significantly associated with the outcomes in many of the countries. Our findings reinforce the importance of the Nurturing Care Framework which describes a multi-sectoral approach to promote ECD in LMICs.

## Introduction

Early childhood is a critical time for the development of children, setting the foundation for their futures. The process of early childhood development (ECD) is complex. It begins at

**Funding:** This study was conducted with support from the United States Agency for International Development (USAID) through The DHS Program (#720-OAA-18C-00083) and the Bill and Melinda Gates Foundation (INV-008034). Under the grant conditions of the Foundation, a Creative Commons Attribution 4.0 Generic License has already been assigned to the Author Accepted Manuscript version that might arise from this submission. EM was involved in the design, analysis, and preparation of the manuscript. The contents are the responsibility of the authors and do not necessarily reflect the views of USAID or the U.S. Government.

**Competing interests:** I have read the journal's policy and the authors of this manuscript have the following competing interests: RKB, TWP, and SR received funding support for this work from United States Agency for International Development (USAID) and RKB received publication support from the Bill and Melinda Gates Foundation. EM is employed through the USAID funded Sustaining Technical and Analytical Resources (STAR) mechanisms and is employed by one of the implementers, The Public Health Institute. The opinions herein are those of the authors and do not necessarily reflect the views of the USAID or the U. S. Government, or the Public Health Institute. Further, this does not alter our adherence to PLOS ONE policies on sharing data and materials.

conception with rapid brain development through age three years and is shaped by stimulation and interaction with social and physical environments, the availability of good nutrition, and other genetic factors [1, 2]. During this process children build their motor, cognitive, social, emotional, language, and self-regulation skills [1, 3]. Children who meet their developmental potential, achieving key developmental milestones, are more likely to continue to increase their learning capacities, achieve academically, and in later life increase their economic productivity.

In 2010 43% of children under the age of five years in low- and middle-income countries (LMICs) were estimated to be at risk of not meeting their developmental potential [4]. Further analyses show that almost one-third of pre-school aged children are not achieving key cognitive and/or social and emotional developmental milestones [5]. Children in LMICs are often exposed to multiple risk factors for poor childhood development [6]. These include structural factors such as poverty and poor environmental conditions, physiological and social factors such as non-responsive caregiving and poor caregiver mental health, and individual level risk factors such as infection and inflammation, stunting, and micronutrient deficiencies [2, 3, 7–10].

Anemia, defined as a low concentration of hemoglobin, is a significant public health problem in several LMICs, especially among young children and women of reproductive age [11]. While anemia can be caused by a variety of factors including micronutrient deficiencies, malaria, infections, chronic inflammation, and genetic disorders, iron deficiency is a major cause [12]. Iron deficiency is estimated to be responsible for over one billion cases of anemia globally and iron deficiency anemia is a top contributor to morbidity in LMICs [13]. Young children in LMICs who are most vulnerable to anemia are also at greatest risk of developmental delays.

For young children, iron is important for tissue oxygen delivery, tissue growth, and brain development [14, 15]. There are sensitive time points in the neonatal, infancy, and toddler periods during which there are higher iron demands to support brain development [15, 16]. Studies examining the links between ECD and iron deficiency anemia in infants and young children have reported that iron deficiency anemia is associated with poor neurodevelopment outcomes including decreased social-emotional, cognitive, and motor development [2, 15, 17]. In addition, iron supplementation interventions among young children demonstrate mixed results with some reviews reporting benefits in cognitive, mental, and motor development among children of varying ages with and without anemia or iron deficiency anemia [18–20]. However, most of these studies focused on children under age two years.

Studies using nationally representative surveys like the Demographic and Health Surveys (DHS) have found significant associations between suboptimal motor, cognitive, and social-emotional development and poor nutritional status [5, 21, 22] and suboptimal literacy-numeracy development and poor dietary intake among children age three to four years [23]. However, few if any studies, have examined associations between anemia status and ECD in population-based surveys. DHS surveys routinely collect hemoglobin data among children under 5 years and since 2011 have collected ECD variables in several LMICs [24]. Leveraging existing DHS surveys that include both ECD variables and hemoglobin data, our analyses examine the association between anemia status and ECD.

## Materials and methods

### Conceptual framework

The conceptual framework for this study is informed by two existing frameworks (Fig 1). The Larson et al. 2017 framework describes a biological pathway linking anemia, diet, nutritional

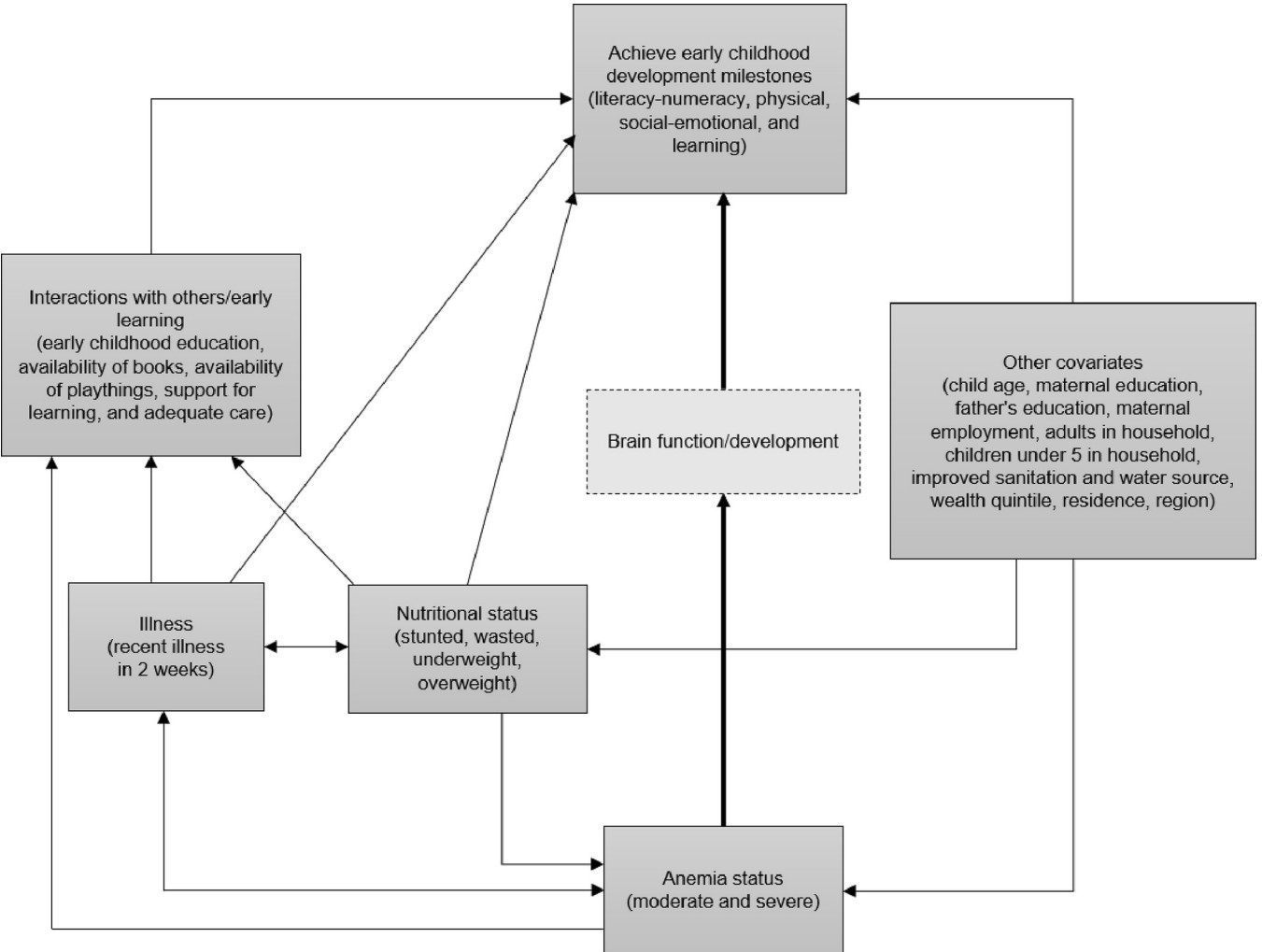

**Fig 1. Conceptual framework of the relationship between anemia status and early childhood development.** Thicker arrow shows pathway that was not directly assessed as brain development data (dashed box) was not available in the datasets. Other arrows and boxes show the pathways and relationships examined in the analyses.

status, illness, and interactions with others and the environment to ECD outcomes [18]. The Nurturing Care Framework takes a social ecological approach [3]. It describes the role of the social context in ECD, with the family environment at the center, and highlights knowledge, attitudes, and behaviors across health, nutrition, security and safety, responsive caregiving, and early learning sectors [2, 3]. The study conceptual framework shows potential pathways between anemia and ECD mediated by brain development as well as potential confounders. Early learning interventions and interactions with others are positively associated with ECD outcomes [2]. Poor nutritional status and child illness directly and indirectly affect ECD and are negatively associated with ECD outcomes [5, 21, 22, 25–28]. Other important confounders include characteristics of the child, mother, father, household, and environment [27, 29].

## Data

Data from nine DHS country surveys were included based on the availability of the ECD questions, anemia testing for children, and recent implementation during the seventh phase of the

**Table 1. Unweighted number of cases in the nine DHS surveys with measurement of combinations of anemia and ECD data for children age 36–59 months.**

| Country | Year | Sample of children 36–59 months | Sample of children 36–59 months with anemia testing data | Sample of children 36–59 months with ECD data | Sample of children 36–59 months with both anemia and ECD data |
|---|---|---|---|---|---|
| Benin | 2017–18 | 4,855 | 2,337 | 4,244 | 2,168 |
| Burundi | 2016–17 | 4,847 | 2,329 | 4,717 | 2,327 |
| Cambodia | 2014 | 2,679 | 1,636 | 2,610 | 1,627 |
| Haiti | 2016–17 | 2,403 | 2,128 | 1,433 | 1,412 |
| Jordan | 2017–18 | 4,184 | 3,877 | 2,137 | 1,996 |
| Maldives | 2016–17 | 1,285 | 907 | 1,264 | 906 |
| Rwanda | 2019–20 | 3,067 | 1,449 | 2,926 | 1,448 |
| Senegal | 2017 | 4,594 | 4,079 | 4,268 | 4,044 |
| Uganda | 2016 | 5,781 | 1,628 | 5,049 | 1,610 |

DHS Program (circa 2013–2019) (Table 1). DHS surveys are population-based household surveys that are representative at the national and the subnational levels. Eligible women age 15–49 years are asked questions about maternal and child health and nutrition, and ECD, among other topics. Biomarkers including anemia (hemoglobin) testing are also collected from eligible women and children. Since children age 6–59 months were tested for anemia and the youngest children age 36–59 months were asked about in the ECD module, the analytical sample is limited to children age 36–59 months (Table 1). Reductions of sample size in some countries are due to subsampling for hemoglobin testing (Benin, Burundi, Cambodia, Rwanda, and Uganda), ECD questions (Jordan and Haiti), and the child not residing in the same household as the mother, which is a requirement for both hemoglobin testing and the ECD indicators in the children's recode dataset. There were significant differences in anemia for children with and without ECD data in Cambodia and significant differences by mean ECD Index score for children with and without anemia data in Haiti, Jordan, and Rwanda (S1 Table). However, these differences were explained by the subsampling for either hemoglobin testing or the ECD questions and the child not residing in the same household as the mother.

## Variables

**Outcome variables.** The outcome variables were measured using the Early Childhood Development Index (ECDI), a validated summary ECD measure based on 10 items responded to by the mother or caregiver of children age 36–59 months [30]. The questions cover four domains of child development: physical, social-emotional, learning, and literacy-numeracy (Table 2). The ECDI was calculated by domain as a binary (yes/no) variable to indicate that a child is on-track in that domain, and overall as a binary (yes/no) variable to indicate that a child is on-track with their overall development.

**Independent variables.** The key independent variable of interest for this study was anemia, which was defined using the WHO recommended cutoffs for hemoglobin concentrations [31]. Children were defined as not anemic when their hemoglobin level was at or above 11.0 grams per deciliter (g/dL), adjusted for altitude in enumeration areas that are above 1,000 meters. Anemia was then categorized into severe (hemoglobin level below 7.0 g/dL), moderate (hemoglobin level between 7.0–9.9 g/dL), and mild (hemoglobin level between 10.0–10.9 g/

**Table 2. ECDI domains, items, and scoring.**

| Domain | Items | On-track score |
|---|---|---|
| Physical | 1. child can pick up small objects with two fingers, like a stick or a rock from the ground; | 2 of 2 items |
| | 2. child is not sometimes too sick to play. | |
| Social-emotional | 3. child gets along well with other children; | 1 of 3 items |
| | 4. child does not kick, bite or hit other children; | |
| | 5. child does not get distracted easily. | |
| Learning | 6. child can follow simple directions on how to do something correctly; | 2 of 2 items |
| | 7. when given something to do, the child is able to do it independently. | |
| Literacy-numeracy | 8. child can read at least four simple, popular words; | 1 of 2 items |
| | 9. can identify/name at least ten letters of the alphabet; | |
| | 10. knows the name and recognizes the symbols of all numbers 1–10. | |
| Overall (ECDI) | | 3 of 4 domains |

dL). Hemoglobin concentrations were measured using capillary blood in the HemoCue® machine [24].

For the purposes of our study, we explored all categorizations of anemia, but for the analyses presented the classification is reduced to 1: severe or moderate; 0: mild or not anemic. Severe or moderate anemia were hypothesized to be more strongly associated with the outcomes based on evidence from studies linking severity of iron deficiency anemia and ECD [15, 32].

Other covariates included attendance at an early childhood education program (yes/no), availability of children's books (child has 3 or more books available; yes/no), availability of playthings (child has at least 2 toys available; yes/no), support for learning (child engaged in 1 or more activities with an adult), adequate care (child was not left alone or in the care of another child less than 10 years of age for more than an hour at any time in the past week; yes/no), child's nutritional status (stunted, underweight, overweight, or wasted; yes/no), wellness status (illness in the past 2 weeks; yes/no), child age, maternal height (short stature; yes/no), maternal work status (worked in last 7 days; yes/no), maternal education (no education/primary education/secondary/higher), paternal education (no education/primary education/secondary/higher), number of adults in the household (less than 3/3 or more), number of children under age five years in the household (less than 3/3 or more), wealth index (5 quintiles), improved sanitation (yes/no), improved water source (yes/no), place of residence, and region. S2 Table provides further details on the definition and categories for the covariates and S3 Table shows the descriptive statistics for the covariates.

## Statistical analysis

We examined the association between anemia status, the overall ECDI, and individual ECDI domains among children age 36–59 months. All analyses were adjusted for the weights and sample design (with strata and clusters) for each survey. Surveys were analyzed separately and not pooled because the relationships differ from one survey to another. The generalized linear models (*glm*) set of estimation commands were used for logit regressions. Model 1 examines the effect of anemia on ECD outcomes with no other covariates. Model 2 adds to Model 1 several child, family, and household covariates: child's age, wellness, and nutritional status; the mother's work status, education, and height; the father's education; the household wealth

quintile; improved water and sanitation; place of residence; and region. Finally Model 3 adds to Model 2 covariates on early learning/interaction variables. Model 2 and Model 3 are important for identifying the effects of anemia on ECD net of the control variables and control plus early learning/interaction variables. Statistical significance was indicated, using $p < 0.05$ criteria. We tested for multicollinearity and found no evidence of collinearity among the variables (S4 Table). The sample sizes varied considerably across the selected surveys, so there was variation in the power of the surveys to detect associations that exist in the reference populations. If a test statistic is not statistically significant, we cannot conclude that there is no association. Stata Version 17 (StataCorp LP, College Station, TX) was used for all statistical analyses.

### Ethical statement

The Institutional Review Board (IRB) of ICF and host country IRBs reviewed and approved the respective DHS surveys. All surveys in this study were conducted with those approvals and survey participants provided verbal consent. The IRB of ICF complied with the United States Department of Health and Human Services regulations for the protection of human research subjects (45 CFR 46). Prior to survey release, all DHS datasets were anonymized.

## Results

### Percentage of children with anemia

More than 40% of children age 36–59 months had any anemia in 7 out of 9 countries and the prevalence of any anemia ranged from 26% in Rwanda to 62% in Benin (Fig 2). The percentage of children with mild anemia ranged from 17% in Rwanda to 31% in Haiti and Cambodia. The prevalence of moderate anemia ranged from 8% in Jordan to 32% in Benin, and the prevalence of severe anemia ranged from less than 1% in Cambodia, Jordan, and Rwanda to 3% in Burundi and Senegal (Fig 2).

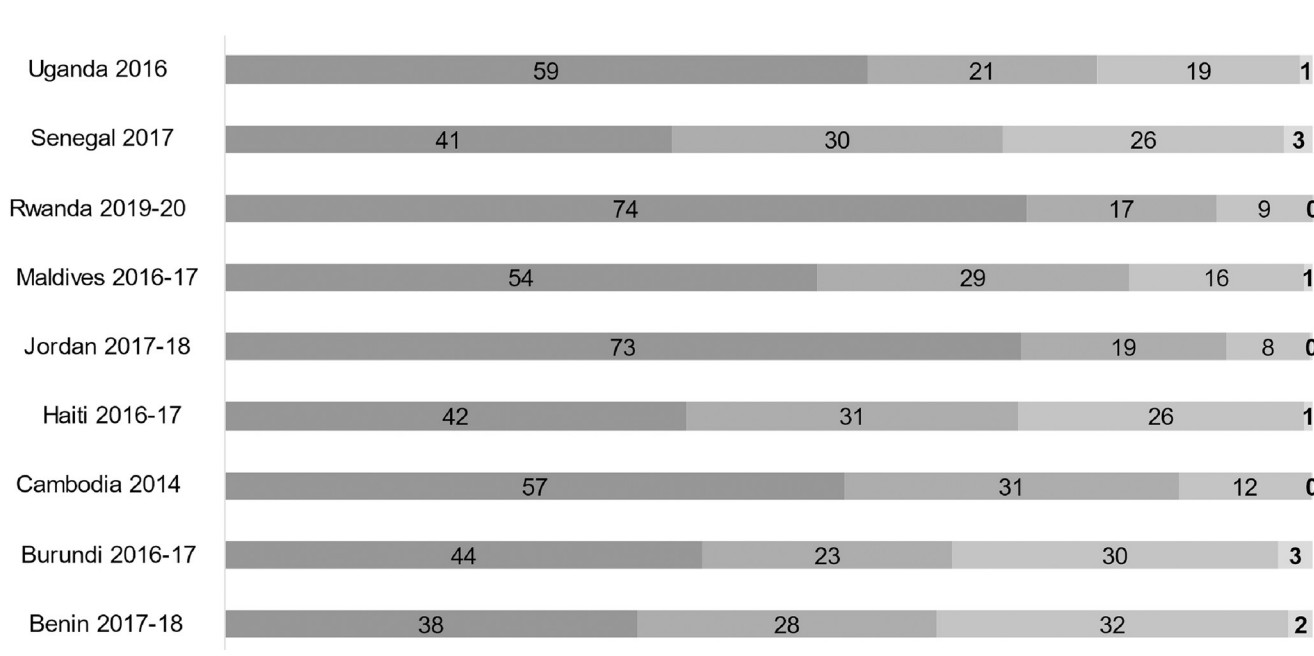

**Fig 2. Percentage of children age 36–59 months with anemia by severity.**

### Early childhood development index and domain scores

The percentage of children who were developmentally on-track for at least three of the four ECDI domains ranged from 41% in Burundi to 93% in Maldives (Fig 3).

Most children were developmentally on-track for the physical domain, ranging from 91% in Uganda to 99% in Maldives (Fig 4). For the social-emotional domain the percentage of children on-track ranged from 60% in Burundi to 95% in Rwanda, and for the learning domain the percentage of children on-track ranged from 63% in Burundi to 96% in Maldives. The literacy-numeracy domain had the lowest percentages of children developmentally on-track overall, with 8 out of 9 countries showing less than 36% of children on track. Percentages ranged from 4% in Senegal to 85% in Maldives (Fig 4).

### Regression results

To make the results easier to interpret, the results for each model were limited to an adjusted odds ratio for the association between anemia and the ECD domains and overall index (Table 3). In Model 1, having moderate or severe anemia was significantly associated with lower odds of being developmentally on-track for several domains and the overall index in many countries, but the magnitudes of the associations were small ranging from an odds ratio (*OR*) of 0.99 to 1.0, and thus may not be clinically meaningful (Table 3). In Benin, Burundi, Haiti, Senegal, and Uganda having moderate or severe anemia was associated with lower odds of on-track literacy-numeracy development (all *ORs* = 0.99 $p < 0.05$). In Benin, Burundi, Jordan, Rwanda, and Senegal, having moderate or severe anemia was associated with lower odds of on-track learning development (*ORs* = 0.99 $p < 0.05$ in Jordan and Rwanda and *ORs* = 1.00 $p < 0.05$ in Benin, Burundi, and Senegal). Having moderate or severe anemia was associated with lower odds of on-track physical development in Haiti and Uganda (*ORs* = 0.99 $p < 0.05$);

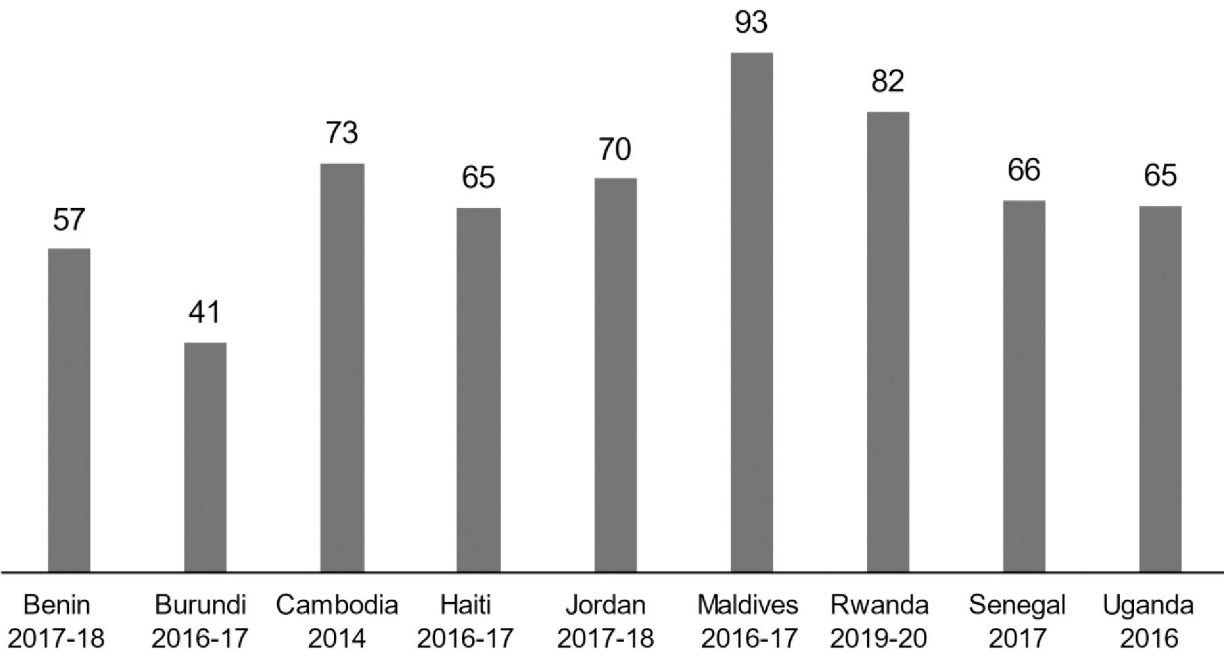

**Fig 3. Percentage of children age 36–59 months developmentally on-track by country.**

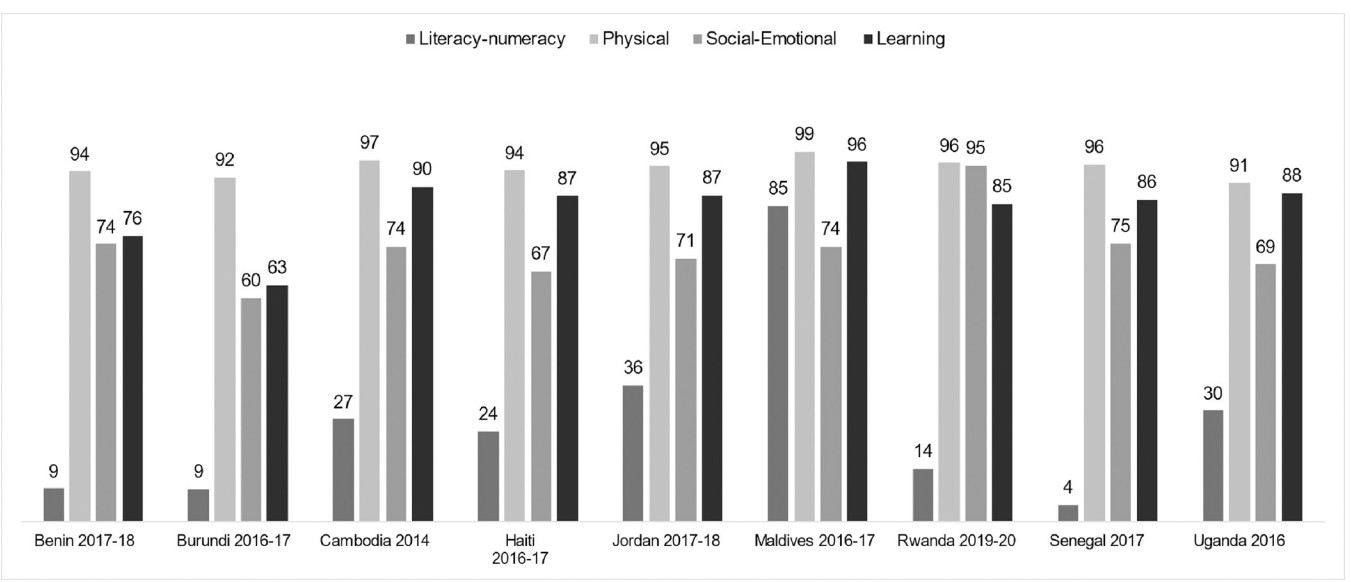

**Fig 4. Percentage of children age 36–59 months on-track for the physical, social-emotional, learning, and literacy-numeracy ECDI domains by country.**

and lower odds of on-track social-emotional development in Benin and Senegal (*ORs* = 1.00 $p$ < 0.05). For the overall index, negative associations among children with moderate or severe anemia were found in Benin, Burundi, Rwanda and Senegal (*OR* = 0.99 $p$ < 0.05 in Rwanda and *OR* = 1.00 $p$ < 0.05 in Benin, Burundi and Senegal) (Table 3).

When adjusted with child, family, and household characteristics (Model 2) and early learning/interaction variables (Model 3), most of the significant findings disappeared (Table 3). In Model 2, there were significant associations between having moderate or severe anemia and lower odds of on-track physical development in Haiti (*AOR* = 0.99 $p$ < 0.05), learning development in Jordan (*AOR* = 0.99 $p$ < 0.05), and social-emotional development and overall index in Benin (*AORs* = 1.00 $p$ < 0.05). With the addition of the early learning/interaction variables to the model (Model 3), having moderate or severe anemia was only associated with lower odds of on-track social-emotional development in Benin (*AOR* = 1.00 $p$ < 0.05) and physical development in Maldives (*AORs* = 0.97 $p$ < 0.05).

In the fully adjusted models (Model 3), several covariates were significantly associated with ECD domains and the overall index, but results varied by country (S5–S9 Tables). Attendance at an early childhood education program was consistently associated with on-track development for the overall ECDI in six out of nine countries, literacy-numeracy in all nine countries, learning in six countries, and social-emotional and physical development in one country each. Small and mostly positive associations were observed between child's age and being on-track for the overall ECDI in five out of nine countries, literacy-numeracy development in eight countries, learning development in four countries, and physical development in two countries. Wealth was associated with being on-track for the overall ECDI, literacy-numeracy, social-emotional, and physical developments in four countries each, and learning development in two countries, but the direction of association varied across countries. For availability of books, playthings, support for learning, adequate care, mother's education, employment, father's education, child nutritional status, wellness, water, sanitation, and residence, significant results were found with some ECD outcomes in fewer countries, but the direction of the associations were not consistent (S5–S9 Tables).

**Table 3. Odds ratios (OR) and adjusted odds ratios (AORs) for the effect of anemia on on-track early childhood development domains and overall index among children age 36–59 months across all countries.**

| Survey | ECD Domain/Index | Model 1[a] | | Model 2[b] | | Model 3[c] | |
|---|---|---|---|---|---|---|---|
| | | OR (95% CI) | p-value | AOR (95% CI) | p-value | AOR (95% CI) | p-value |
| Benin 2017–18 | Literacy-numeracy | 0.99 (0.99–1.00) | 0.004 | 1.00 (0.99–1.01) | 0.880 | 1.00 (0.99–1.01) | 0.924 |
| | Physical | 1.00 (1.00–1.00) | 0.390 | 1.00 (1.00–1.01) | 0.354 | 1.00 (1.00–1.01) | 0.358 |
| | Social-emotional | 1.00 (1.00–1.00) | 0.042 | 1.00 (0.99–1.00) | 0.021 | 1.00 (0.99–1.00) | 0.023 |
| | Learning | 1.00 (0.99–1.00) | 0.001 | 1.00 (1.00–1.00) | 0.507 | 1.00 (1.00–1.00) | 0.637 |
| | ECDI | 1.00 (0.99–1.00) | 0.000 | 1.00 (1.00–1.00) | 0.043 | 1.00 (1.00–1.00) | 0.063 |
| Burundi 2016–17 | Literacy-numeracy | 0.99 (0.99–1.00) | 0.005 | 1.00 (0.99–1.00) | 0.722 | 1.00 (1.00–1.01) | 0.715 |
| | Physical | 1.00 (0.99–1.00) | 0.165 | 1.00 (0.99–1.00) | 0.853 | 1.00 (0.99–1.00) | 0.855 |
| | Social-emotional | 1.00 (1.00–1.00) | 0.258 | 1.00 (1.00–1.00) | 0.549 | 1.00 (1.00–1.00) | 0.469 |
| | Learning | 1.00 (1.00–1.00) | 0.030 | 1.00 (1.00–1.00) | 0.360 | 1.00 (1.00–1.00) | 0.482 |
| | ECDI | 1.00 (1.00–1.00) | 0.007 | 1.00 (1.00–1.00) | 0.175 | 1.00 (1.00–1.00) | 0.353 |
| Cambodia 2014 | Literacy-numeracy | 1.00 (0.99–1.00) | 0.343 | 1.00 (1.00–1.01) | 0.723 | 1.00 (1.00–1.01) | 0.487 |
| | Physical | 0.99 (0.99–1.00) | 0.260 | 0.99 (0.98–1.00) | 0.100 | 0.99 (0.98–1.00) | 0.099 |
| | Social-emotional | 1.00 (0.99–1.00) | 0.541 | 1.00 (0.99–1.00) | 0.227 | 1.00 (0.99–1.00) | 0.261 |
| | Learning | 1.00 (0.99–1.01) | 0.938 | 1.00 (1.00–1.01) | 0.751 | 1.00 (1.00–1.01) | 0.662 |
| | ECDI | 1.00 (0.99–1.00) | 0.337 | 1.00 (0.99–1.00) | 0.447 | 1.00 (0.99–1.00) | 0.548 |
| Haiti 2016–17 | Literacy-numeracy | 0.99 (0.99–1.00) | 0.000 | 0.99 (0.99–1.00) | 0.101 | 0.99 (0.99–1.00) | 0.133 |
| | Physical | 0.99 (0.98–1.00) | 0.001 | 0.99 (0.98–1.00) | 0.038 | 0.99 (0.98–1.00) | 0.052 |
| | Social-emotional | 1.00 (1.00–1.00) | 0.796 | 1.00 (1.00–1.01) | 0.602 | 1.00 (1.00–1.01) | 0.465 |
| | Learning | 1.00 (1.00–1.00) | 0.832 | 1.00 (1.00–1.01) | 0.515 | 1.00 (1.00–1.01) | 0.374 |
| | ECDI | 1.00 (0.99–1.00) | 0.129 | 1.00 (1.00–1.01) | 0.543 | 1.00 (1.00–1.01) | 0.393 |
| Jordan 2017–18 | Literacy-numeracy | 1.00 (0.99–1.00) | 0.102 | 1.00 (0.99–1.00) | 0.490 | 1.00 (0.99–1.00) | 0.547 |
| | Physical | 1.00 (0.98–1.01) | 0.745 | 1.00 (0.98–1.01) | 0.707 | 1.00 (0.98–1.01) | 0.686 |
| | Social-emotional | 1.00 (1.00–1.01) | 0.286 | 1.00 (1.00–1.01) | 0.480 | 1.00 (1.00–1.01) | 0.428 |
| | Learning | 0.99 (0.99–1.00) | 0.011 | 0.99 (0.99–1.00) | 0.041 | 0.99 (0.99–1.00) | 0.100 |
| | ECDI | 1.00 (0.99–1.00) | 0.064 | 1.00 (0.99–1.00) | 0.155 | 1.00 (0.99–1.00) | 0.215 |
| Maldives 2016–17 | Literacy-numeracy | 1.00 (0.99–1.00) | 0.466 | 1.00 (0.98–1.01) | 0.603 | 1.00 (0.98–1.01) | 0.532 |
| | Physical | 0.99 (0.98–1.00) | 0.220 | 0.98 (0.95–1.00) | 0.095 | 0.97 (0.95–0.98) | 0.001 |
| | Social-emotional | 1.00 (0.99–1.01) | 0.746 | 1.00 (0.99–1.00) | 0.281 | 1.00 (0.99–1.01) | 0.565 |
| | Learning | 1.01 (1.00–1.02) | 0.208 | 1.01 (0.99–1.02) | 0.436 | 1.00 (0.99–1.02) | 0.422 |
| | ECDI | 1.00 (0.99–1.01) | 0.996 | 1.00 (0.99–1.02) | 0.480 | 1.01 (0.99–1.02) | 0.330 |
| Rwanda 2019–20 | Literacy-numeracy | 1.00 (0.99–1.00) | 0.541 | 1.00 (0.99–1.01) | 0.690 | 1.00 (0.99–1.01) | 0.997 |
| | Physical | 1.01 (1.00–1.02) | 0.140 | 1.01 (1.00–1.03) | 0.124 | 1.01 (1.00–1.03) | 0.120 |
| | Social-emotional | 1.00 (0.99–1.00) | 0.366 | 1.00 (0.99–1.01) | 0.829 | 1.00 (0.99–1.01) | 0.741 |
| | Learning | 0.99 (0.99–1.00) | 0.007 | 1.00 (0.99–1.00) | 0.472 | 1.00 (0.99–1.00) | 0.328 |
| | ECDI | 0.99 (0.99–1.00) | 0.027 | 1.00 (1.00–1.01) | 0.913 | 1.00 (0.99–1.00) | 0.889 |
| Senegal 2017 | Literacy-numeracy | 0.99 (0.98–0.99) | 0.000 | 1.00 (0.99–1.00) | 0.146 | 1.00 (0.99–1.00) | 0.419 |
| | Physical | 1.00 (0.99–1.00) | 0.254 | 1.00 (0.99–1.00) | 0.814 | 1.00 (0.99–1.00) | 0.402 |
| | Social-emotional | 1.00 (1.00–1.00) | 0.031 | 1.00 (1.00–1.00) | 0.744 | 1.00 (1.00–1.00) | 0.421 |
| | Learning | 1.00 (0.99–1.00) | 0.000 | 1.00 (1.00–1.00) | 0.567 | 1.00 (1.00–1.00) | 0.649 |
| | ECDI | 1.00 (0.99–1.00) | 0.000 | 1.00 (1.00–1.00) | 0.441 | 1.00 (1.00–1.00) | 0.297 |
| Uganda 2016 | Literacy-numeracy | 0.99 (0.99–1.00) | 0.000 | 1.00 (1.00–1.01) | 0.578 | 1.00 (1.00–1.01) | 0.316 |
| | Physical | 0.99 (0.99–1.00) | 0.000 | 1.00 (0.99–1.00) | 0.400 | 1.00 (0.99–1.00) | 0.294 |
| | Social-emotional | 1.00 (1.00–1.00) | 0.439 | 1.00 (1.00–1.01) | 0.154 | 1.00 (1.00–1.01) | 0.173 |
| | Learning | 1.00 (0.99–1.00) | 0.463 | 1.00 (1.00–1.01) | 0.229 | 1.00 (1.00–1.01) | 0.242 |

*(Continued)*

**Table 3.** (Continued)

| Survey | ECD Domain/Index | Model 1[a] | | Model 2[b] | | Model 3[c] | |
|---|---|---|---|---|---|---|---|
| | | OR (95% CI) | p-value | AOR (95% CI) | p-value | AOR (95% CI) | p-value |
| | ECDI | 1.00 (0.99–1.00) | 0.091 | 1.00 (1.00–1.01) | 0.114 | 1.00 (1.00–1.01) | 0.099 |

OR = odds ratio, AOR = adjusted odds ratio, LB = Lower bound of 95% confidence interval; UB = upper bound of 95% confidence interval

[a]Model 1: bivariate model with child anemia (moderate or severe)

[b]Model 2: adjusted for child nutritional status, child wellness, age of child, maternal height, education, employment, paternal education, number of adults age 15+ in household, number of children under age 5 in household, improved sanitation, improved water source, wealth index, residence, region. In Jordan, child nutritional status was not included and in Senegal, maternal height was not included

[c]Model 3: same as model 2 covariates plus early childhood education attendance, availability of books, availability of playthings, support for learning, and adequate care

## Discussion

This study examined associations between child anemia and ECD outcomes in population-based surveys in nine LMICs. In most of the countries, anemia was a serious public health problem, and more than half of children were developmentally on-track, but there was variation across countries. Significant associations with several developmental outcomes were observed in bivariate analyses, but ORs were close to one. In adjusted analyses, most associations were attenuated. There were only two statistically significant findings with social-emotional and physical development in Benin and Maldives, respectively. Both associations were small.

The unadjusted association between having anemia and ECD showed that children with moderate or severe anemia were less likely to be on-track developmentally for literacy-numeracy, physical, social-emotional, learning, and the overall index in several countries. However, these associations were very small, may have limited clinical significance, and it is also plausible that some may be spurious. Nevertheless, the direction of the associations were consistent for all developmental domains and the index across all countries. Our finding agrees with our hypothesized pathway between anemia and ECD and is supported by studies showing poorer cognitive, motor, and social-emotional development associated with iron deficiency anemia in young children under age two years in LMICs [15]. Further, other studies have reported significant associations with modest effect sizes between higher hemoglobin concentrations and better language, motor, and cognitive development in children under two years in LMICs [27, 28]. Given that we used moderate or severe anemia status as a proxy for iron deficiency anemia in our study, the limited significant associations and ORs very close to 1 are not surprising. Furthermore, it is possible for children to be iron deficient without anemia and this may also account for the small associations observed.

In adjusted regressions, we found a lack of association between anemia and ECD outcomes except for small associations with social-emotional and physical development in two countries. While this finding is contrary to other studies from baseline or endline assessments of micronutrient and lipid supplement interventions that show a complex relationship between anemia and ECD among children under two years, there are many limitations to these analyses and differences with such research that contributes to such disparities (e.g. only looking at cross-sectional data among children age 36–59 months, ECD assessment method, sample size) [25, 27, 28]. In Ghana and Malawi, authors conducted path analyses and reported weak but significant associations between hemoglobin/iron status and language development among children age 6–18 months, and stronger (direct) associations between hemoglobin/iron status and motor development among children age 9–18 months in Burkina Faso when controlling for child, caregiver, and household confounders [27]. Among Zanzibari children age 15–19

months, path analyses revealed significant associations between hemoglobin concentration and motor development, and in India, weak associations between hemoglobin concentration, language, social, and cognitive development were reported among children age 12–18 months [25, 28].

There are a few explanations for our different findings. Compared to our study, researchers in previous studies used different and more direct and specific measures of ECD such as the Developmental Milestones Checklist-II, vocabulary checklists, and task tests which may partly explain our null results [25, 27, 28]. Such assessments involve interviews and direct observations for a variety of domains based on age-specific milestones [33–36]. In addition, the prior studies used data from supplement trials and children who received the interventions may be more likely to show associations with ECD outcomes. Another reason is that anemia generally peaks around 12–24 months of age, which is a sensitive time for brain development and most studies examining the effects of iron deficiency anemia have focused on children under two years [15, 25, 27, 28]. Brain development is most abundant during the first three years of life and therefore, assessing children age 36–59 months may not represent the most sensitive period to find associations [1, 37]. By exploring relationships among older children, we may have potentially missed the biologically relevant window and thus our analyses find no associations. Another potential explanation is that we assessed current anemia status which may not reflect chronicity. The duration of anemia/iron deficiency anemia has been associated with worse cognitive and motor development in young children in Chile and Guatemala [15]. The effects of anemia on ECD may not be immediate and these cross-sectional analyses do not allow for understanding whether anemia was experienced before developmental delays.

Our analyses also controlled for several covariates highlighting the importance of early childhood education, age in months, and wealth status on ECD. Our finding that in most countries, attendance at an early childhood education program was strongly associated with on-track ECD outcomes is consistent with other evidence among young children in LMICs (S2–S6 Tables). Two reviews found positive effects of early childhood education programs on literacy, psychosocial, and other cognitive development measures among preschool aged children and children under age two years in LMICs [38, 39] and recent studies found similar results for language, psychosocial, and motor development among young children ranging from age 12–59 months in LMICs [23, 27, 28]. We also observed small positive associations with age and in general children from wealthier households were more likely to be on-track developmentally (S2–S6 Tables). Both variables likely have some interaction with early childhood education, but findings for wealth are consistent with others showing disparities in ECD by socioeconomic status [27, 38]. Overall, as evidenced by the Nurturing Care Framework, many factors contribute to a child's development and anemia is only one potential adverse exposure a child could have.

## Strengths and limitations

To our knowledge this was the first study to examine the association between anemia and ECD using population-based survey data in LMICs. Although our results did not show strong or consistent results across countries, they are representative of entire populations and the adjusted models reinforce the importance of the multi-sectoral Nurturing Care Framework for policy and programmatic approaches that address country-specific social and environmental contexts.

Our study is not without limitations. The cross-sectional nature of the data means we cannot infer causality and thus the direction of associations is not certain. In addition, there is potential of selection bias in the analytical sample since the children all resided in the same household as the mother and they could differ from children not living with their mother.

However, the consistency of our few results with other studies on the direction of association provides some reassurance. In some countries there may have been limited power to detect associations because of sample size. Pooling would have alleviated this, but settings differed and there could be setting-specific effects on measurements. We already discussed several other challenges related to the key independent variable of interest, i.e. anemia as a proxy, representing current status, and temporality and how this may have contributed to the null findings and weak associations. In addition, anemia was calculated as a binary variable. We could have used hemoglobin concentration as a continuous variable but the results were similar (results not shown). Additionally, we considered using only severe anemia, but that would have limited our sample size. Inclusion of markers of iron deficiency could have strengthened the analysis, but these are not collected in DHS surveys. We controlled for several confounders based on conceptual relevance because we wanted to examine the effect of anemia in context, but these variables likely attenuated our already weak associations. Path or mediation analyses could be conducted to better understand the pathways and examine direct and indirect contributions of anemia. We examined pairwise relationships between variables in Fig 1 for all surveys (results not shown). Several unadjusted odds ratios for these paths were significant, but after adjusting for the control variables listed above, very few significant paths remained. In addition, there could be unaccounted factors such as different domestic policies on ECD or inequities in healthcare access that could impact the associations between countries. Another limitation is our outcome. Although the ECDI has been validated for the age group we analyzed, the literacy-numeracy and physical domains of the ECDI have been criticized [1, 5, 23]. The literacy-numeracy domain of the ECDI may include items that are too advanced for children age 36–59 months, and the physical domain of the ECDI may include items that are basic functionalities for most children age 36–59 months [5]. Another criticism is that the ECDI is missing domains or items that capture other relevant cognitive functions which develop between 36–59 months such as increased attention span and processing speed which are more strongly linked to iron status [17, 23]. The ECDI was also not intended to be used for domain-specific analyses. The new ECDI 2030 overcomes many of the challenges with the ECDI, but at the time of these surveys was not available. The ECDI is also based on caregiver recall and therefore is subject to recall bias and not as rigorous as direct child observations.

## Conclusion

Our study found few associations between severe or moderate anemia and ECD domains and the overall ECD index among children age 36–59 months. Positive associations were observed for early learning/interaction variables and a few demographic variables. Multi-sectoral interventions that target children early and address disparities are important to promote ECD in LMICs. Future research could replicate these analyses with the new ECDI 2030 or direct developmental assessments which address some of the limitations of the ECDI, and longitudinal studies with younger children could also address temporality issues. Additionally, studies unpacking the complex pathways between nutrition indicators and all domains of nurturing care, and early childhood development outcomes in different contexts could inform targeted ECD policies and programs. Alone, effects of nutrition and health interventions on ECD can be modest, but in tandem with interventions from other sectors, they can contribute to promoting optimal ECD.

## Supporting information

**S1 Table. Percentage of children with and without ECD data among children with any anemia and the percentage of children with and without anemia among children's mean ECD**

**index.**
(DOCX)

**S2 Table. Variables used in analysis.** Notes: Maternal height was not collected in Senegal 2017. Child nutritional status was not included in Jordan 2017–18 due to data quality concerns. [1] Stunted, underweight, overweight, or wasted were categorized as follows: children under 5 in the household were categorized as underweight, or normal according to the weight-for-age Z-score, categorized as stunted or normal according to the height-for-age Z-score, and categorized as wasted, normal, or overweight according to the weight-for-height Z-score in comparison to the mean on the WHO Child Growth Standards scale.
(DOCX)

**S3 Table. Descriptive statistics of variables, stratified by anemia status for each country.** In the descriptive table, education was categorized as any education versus none; wealth was categorized as top 3 wealth quintiles versus the bottom 2 quintiles; and WASH was categorized as improved water and sanitation versus other. [+]Malnourished refers to children who are not stunted, underweight, overweight, or wasted.
(DOCX)

**S4 Table. Variance inflation factors (VIF) for covariates included in the regression for each country.** In the table, education was categorized as any education versus none; wealth was categorized as top 3 wealth quintiles versus the bottom 2 quintiles; and WASH as categorized as improved water and sanitation versus other. [+]Malnourished refers to children who are not stunted, underweight, overweight, or wasted.
(DOCX)

**S5 Table. Adjusted odds ratios for on-track early childhood development among children age 36–59 months across all countries.** AOR = adjusted odds ratio, LB = Lower bound of 95% confidence interval; UB = upper bound of 95% confidence interval. Models were adjusted for region. In Jordan child nutritional status was not included and in Senegal maternal height was not included. Blank cells indicate that no coefficients were produced in the model because of small sample sizes. [+] Malnourished refers to children who are not stunted, underweight, overweight, or wasted.
(DOCX)

**S6 Table. Adjusted odds ratios for on-track literacy-numeracy development among children age 36–59 months across all countries.** AOR = adjusted odds ratio, LB = Lower bound of 95% confidence interval; UB = upper bound of 95% confidence interval. Models were adjusted for region. In Jordan child nutritional status was not included and in Senegal maternal height was not included. Blank cells indicate that no coefficients were produced in the model because of small sample sizes. [+] Malnourished refers to children who are not stunted, underweight, overweight, or wasted.
(DOCX)

**S7 Table. Adjusted odds ratios for on-track physical development among children age 36–59 months across all countries.** AOR = adjusted odds ratio, LB = Lower bound of 95% confidence interval; UB = upper bound of 95% confidence interval. Models were adjusted for region. In Jordan child nutritional status was not included and in Senegal maternal height was not included. Blank cells indicate that no coefficients were produced in the model because of small sample sizes. [+] Malnourished refers to children who are not stunted, underweight, overweight, or wasted.
(DOCX)

**S8 Table. Adjusted odds ratios for on-track for social-emotional development among children age 36–59 months across all countries.** AOR = adjusted odds ratio, LB = Lower bound of 95% confidence interval; UB = upper bound of 95% confidence interval. Models were adjusted for region. In Jordan child nutritional status was not included and in Senegal maternal height was not included. Blank cells indicate that no coefficients were produced in the model because of small sample sizes. [+] Malnourished refers to children who are not stunted, underweight, overweight, or wasted.
(DOCX)

**S9 Table. Adjusted odds ratios for on-track learning development among children age 36–59 months across all countries.** AOR = adjusted odds ratio, LB = Lower bound of 95% confidence interval; UB = upper bound of 95% confidence interval. Models were adjusted for region. In Jordan child nutritional status was not included and in Senegal maternal height was not included. Blank cells indicate that no coefficients were produced in the model because of small sample sizes. [+] Malnourished refers to children who are not stunted, underweight, overweight, or wasted.
(DOCX)

## Author Contributions

**Conceptualization:** Rukundo K. Benedict, Thomas W. Pullum, Sara Riese, Erin Milner.

**Data curation:** Sara Riese.

**Formal analysis:** Thomas W. Pullum.

**Methodology:** Rukundo K. Benedict, Thomas W. Pullum, Sara Riese, Erin Milner.

**Visualization:** Rukundo K. Benedict.

**Writing – original draft:** Rukundo K. Benedict.

**Writing – review & editing:** Thomas W. Pullum, Sara Riese, Erin Milner.

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
