## [Decision Letter · Decision Letter 0]

6 Mar 2023

PONE-D-23-01169Is child anemia associated with early childhood development? A cross-sectional analysis of nine Demographic and Health SurveysPLOS ONE

Dear Dr. Benedict,

Thank you for submitting your manuscript to PLOS ONE. After careful consideration, we feel that it has merit but does not fully meet PLOS ONE’s publication criteria as it currently stands. Therefore, we invite you to submit a revised version of the manuscript that addresses the points raised during the review process.

We look forward to receiving your revised manuscript.

Kind regards,

Kannan Navaneetham, PhD

Academic Editor

PLOS ONE

Journal Requirements:

3. Thank you for stating the following in the Competing Interest section: 

"I have read the journal's policy and the authors of this manuscript have the following competing interests: Erin Milner is employed through the USAID funded Sustaining Technical and Analytical Resources (STAR) mechanisms and is employed by one of the implementers, The Public Health Institute. The opinions herein are those of the authors and do not necessarily reflect the views of the USAID or the U.S. Government, or the Public Health Institute."

We note that you received funding from a commercial source: USAID

Within this Competing Interests Statement, please confirm that this does not alter your adherence to all PLOS ONE policies on sharing data and materials by including the following statement: ""This does not alter our adherence to PLOS ONE policies on sharing data and materials.” (as detailed online in our guide for authors http://journals.plos.org/plosone/s/competing-interests).  If there are restrictions on sharing of data and/or materials, please state these. Please note that we cannot proceed with consideration of your article until this information has been declared. 

Reviewers' comments:

Reviewer's Responses to Questions

**Comments to the Author**

1. Is the manuscript technically sound, and do the data support the conclusions?

Reviewer #1: Yes

Reviewer #2: Partly

2. Has the statistical analysis been performed appropriately and rigorously? 

Reviewer #1: Yes

Reviewer #2: Yes

3. Have the authors made all data underlying the findings in their manuscript fully available?

Reviewer #1: Yes

Reviewer #2: Yes

4. Is the manuscript presented in an intelligible fashion and written in standard English?

Reviewer #1: Yes

Reviewer #2: Yes

5. Review Comments to the Author

Reviewer #1: This manuscript is well-written and regards an important issue in public health: anemia and early childhood development. The work is even more important because the context includes LMICs, where the risk of both anemia and poor child development outcomes are substantial. Further, the data are from the latest phase of The DHS Program, lending credibility to the results.

It's too bad that the findings were mostly statistically null (and questionably meaningful in the few cases of statistical significance), but the authors do a great job of clearly describing the results without over-interpretation. Further, null findings make important contributions to the scientific literature and to our collective understanding of the complex relationships between nutritional status and brain and cognitive development. The authors should be commended on their clear and thorough description of the likely explanation for their findings, and how the results are not counter to the existing literature. I am left to wonder, however, why the authors did not examine path or mediation analyses, as mentioned on Lines 326-327? It might further support the points made in the discussion about the potential causes of the null findings and weak associations. I am curious if such an analysis might reveal a different angle to this story: do the influences of the environment and caregiver involvement explain (most of) the variation in ECD in these populations, DESPITE anemia? While we should address anemia for a number of reasons, perhaps part of the story here is about the benefits for ECD of investing in Early Childhood Education and supporting caregivers to promote high-quality caregiver-child interactions. I recognize these additional analyses would not remove the limits of cross-sectional data, using anemia as a proxy for ID(A), or of a broad measure of development like the ECDI. But it might allow you to tell a more nuanced story, and to further support the points made in the discussion with the existing data.

Without the mediation analyses, these results are still worth publishing, but I'd love to see a revised version of this manuscript that includes such analyses, and which would have the potential to be more impactful than the current draft. I selected "Major revision" not because I think this is a poor manuscript--on the contrary, I think it's a valuable manuscript. But I know it is no small task to complete mediation analyses, and I only request it here because I believe the addition will enhance the story the authors can tell with these data.

Please see a few minor comments/edits below:

Table 1: Is there anything different about the rates of anemia by country among children who had ECD data vs the full sample? And is there anything different about the ECD scores among children with anemia data vs the null sample? I'm trying to get an idea about potential selection bias in these analyses.

Line 145-146 (and related content in S1 Table): What's the reason for the categorization of <3 vs 3+ people >15y or children <5 in these two variables?

Statistical Analysis: There are a lot of covariates here. Please confirm that you checked assumptions of independence/colinearity, especially with anemia (It's fine to put this in Supplemental materials).

Line 161: Please clarify in which circumstances each p-value cut-off was applied.

Line 198: Please change "the effect of anemia on the ECD..." to "the association between anemia and ECD..." to clarify that these data cannot denote causality.

Line 333-334: This is an important point. It may be worth noting that these domains (attention span and processing speed) have been more strongly linked to iron status than the broader domains included in the ECDI.

Supplemental Tables: Could you please add a table with descriptive statistics of each of the covariates by country (ideally for the total population by country and by anemia status)? It would be helpful for interpreting the regression results in the rest of the supplementation tables.

Reviewer #2: The manuscript was well-written and addressed the concerns of nutrition and child development in LMICs.

Several suggestions to improve the manuscript are listed below:

- What is the country selection based on?

- How do the authors respond to the possible diversity in each country’s domestic policy and regional condition, and other than variables included as covariates in the analysis, are there any possible factors that may cause the variation in the model analysis between countries?

-Figure 1: Please add descriptions for different arrow thicknesses and the different box outlines.

6. PLOS authors have the option to publish the peer review history of their article (what does this mean?). If published, this will include your full peer review and any attached files.

Reviewer #1: No

Reviewer #2: No

---

## [Author Response · Author response to Decision Letter 0]

29 Dec 2023

Please find responses to the reviewer comments. 

Reviewer #1: This manuscript is well-written and regards an important issue in public health: anemia and early childhood development. The work is even more important because the context includes LMICs, where the risk of both anemia and poor child development outcomes are substantial. Further, the data are from the latest phase of The DHS Program, lending credibility to the results.

It's too bad that the findings were mostly statistically null (and questionably meaningful in the few cases of statistical significance), but the authors do a great job of clearly describing the results without over-interpretation. Further, null findings make important contributions to the scientific literature and to our collective understanding of the complex relationships between nutritional status and brain and cognitive development. The authors should be commended on their clear and thorough description of the likely explanation for their findings, and how the results are not counter to the existing literature. I am left to wonder, however, why the authors did not examine path or mediation analyses, as mentioned on Lines 326-327? It might further support the points made in the discussion about the potential causes of the null findings and weak associations. I am curious if such an analysis might reveal a different angle to this story: do the influences of the environment and caregiver involvement explain (most of) the variation in ECD in these populations, DESPITE anemia? While we should address anemia for a number of reasons, perhaps part of the story here is about the benefits for ECD of investing in Early Childhood Education and supporting caregivers to promote high-quality caregiver-child interactions. I recognize these additional analyses would not remove the limits of cross-sectional data, using anemia as a proxy for ID(A), or of a broad measure of development like the ECDI. But it might allow you to tell a more nuanced story, and to further support the points made in the discussion with the existing data.

Without the mediation analyses, these results are still worth publishing, but I'd love to see a revised version of this manuscript that includes such analyses, and which would have the potential to be more impactful than the current draft. I selected "Major revision" not because I think this is a poor manuscript--on the contrary, I think it's a valuable manuscript. But I know it is no small task to complete mediation analyses, and I only request it here because I believe the addition will enhance the story the authors can tell with these data.

Response: Thank you for the suggestion, we fully considered this request but were unable to proceed with the mediation analyses. However, we explored a systematic stepwise “path analysis” using 27 logit pairwise logit regression corresponding to the paths in Figure 1. We started with Benin (since this country showed the most evidence of potential associations between child anemia and ECD outcomes) and looked at the direct and indirect pathways connecting the variables in conceptual model. We did this in three steps. The first step estimates the direct effect of child anemia on the ECD outcomes, ignoring the background or exogenous variables and ignoring the potential pathways hypothesized in the model. The second step, which is conditional on the evidence of direct effects in the first step, includes statistical controls for the background variables. The third step, which is conditional on the results in the second step, articulates the pathways through the mediating variables in the model.

Step 1. The five early learning/interaction variables (which were associated with the ECD outcomes) are as follows:

• Early childhood education 

• Availability of books 

• Availability of playthings 

• Adequate care 

• Support for learning 

One at a time, the early learning/interaction variables are regressed on the binary anemia variable, using logit regression with survey adjustments, to estimate the direct effect of anemia on the early learning/interaction variable. We find a highly significant negative effect of child anemia on the early learning/interaction variables except for adequate care. The p-values are as follows: 

• Early childhood education, p<.001 

• Availability of books, p<.001

• Availability of playthings, p<.001 

• Adequate care, p=0.647

• Support for learning, p<.001 

Step 2. For the early learning/interaction variables other than adequate care (because it did not show a significant relationship in step 1), we re-assessed the direct effect of child anemia on the variables by adding the controls listed earlier (child age, the mother’s education, the father’s education, whether the mother is working, whether the mother is short, whether there are 3+ adults in the household, whether there are 3+ children under 5 in the household, wealth quintile, whether the household has improved water and sanitation, and the country’s regional classification). For all the early learning/interaction variables, the negative relationship found in step 1 disappears. The p-values for the effect of child anemia on the variables is as follows: 

• Early childhood education, p=0.977

• Availability of books, p=0.715

• Availability of playthings, p=0.613

• Support for learning, p=0.793

These p values do not approach any of the standard criteria of statistical significance (p<0.05).

Step 3. If any of the direct effects in step 2 had been significant, we would next examine indirect pathways in the model. However, there is no motivation for such an analysis. We repeated the same analyses for each of the countries and observed very few significant associations at step three. Hence, we did not pursue path analysis further. 

Reviewer comment: Table 1: Is there anything different about the rates of anemia by country among children who had ECD data vs the full sample? And is there anything different about the ECD scores among children with anemia data vs the null sample? I'm trying to get an idea about potential selection bias in these analyses.

Response: We added supplementary Table S1 to show this information and included text int eh data section (lines 111-116). There were no significant differences by anemia for children with or without ECD data except for Cambodia and no differences by mean ECD Index for children with or without anemia data except for Haiti, Jordan, and Rwanda surveys. Further examination found that subsampling for hemoglobin testing or ECD questions explained some of the difference and the rest was explained by children not residing in the same household as the mother (which is a requirement for both hemoglobin testing and the ECD questions in the dataset). It is possible that children not residing within the same household as the mother differ from those residing in the same household as the mother. Therefore, the generalizability of the findings may be limited and we have included this in the limitations section (Lines 333-335).

Reviewer comment: Line 145-146 (and related content in S1 Table): What's the reason for the categorization of <3 vs 3+ people >15y or children <5 in these two variables? 

Response: The categorization assumed two caregivers and another adult in the household. An adult may be an older sibling or may be unrelated to the reference child but is a potential source of interaction with the child. We used the same categorization for children under 5 in the household.

Reviewer comment: Statistical Analysis: There are a lot of covariates here. Please confirm that you checked assumptions of independence/colinearity, especially with anemia (It's fine to put this in Supplemental materials).

Response: Thank you, yes, we can confirm that we checked for collinearity among the covariates. We included language in the statistical analysis section (lines 169-170) and included Supplementary Table S4 which shows the results from the test of multicollinearity. We used the Variance Inflation Factor (VIF) to test for collinearity among the 17 covariates including anemia. VIFs that exceed 4 are generally considered to indicate collinearity. In the table, the values are close to 1, providing no evidence of collinearity.

Reviewer comment: Line 161: Please clarify in which circumstances each p-value cut-off was applied.

Response: Thank you, we have clarified that the p-value <0.05 was used to denote statistical significance in the regressions.

Reviewer comment: Line 198: Please change "the effect of anemia on the ECD..." to "the association between anemia and ECD..." to clarify that these data cannot denote causality.

Response: Thank you for catching this. We have updated the text as indicated.

Reviewer comment: Line 333-334: This is an important point. It may be worth noting that these domains (attention span and processing speed) have been more strongly linked to iron status than the broader domains included in the ECDI.

Response: Thank you, we have edited the text to make this point (line 356)

Reviewer comment: Supplemental Tables: Could you please add a table with descriptive statistics of each of the covariates by country (ideally for the total population by country and by anemia status)? It would be helpful for interpreting the regression results in the rest of the supplementation tables.

Response: We have added this table to the supplementary tables (table S2) and included in the text (line 164).

Reviewer comment: Reviewer #2: The manuscript was well-written and addressed the concerns of nutrition and child development in LMICs.

Several suggestions to improve the manuscript are listed below:

- What is the country selection based on?

Response: The selection was based upon countries with available DHS data. In the text we explain: Data from nine DHS country surveys were included based on the availability of the ECD questions, anemia testing for children, and recent implementation during the seventh phase of the DHS Program (circa 2013-2019) (Table 1).

Reviewer comment: How do the authors respond to the possible diversity in each country’s domestic policy and regional condition, and other than variables included as covariates in the analysis, are there any possible factors that may cause the variation in the model analysis between countries?

Response: Thanks for this excellent question! We examined each country separately, because we thought that there could be country specific factors, beyond what we controlled for that could impact the relationship under investigation (lines 169-170). Certainly, different domestic policies around early childhood education could affect the association as could settings with high endemicity of malaria, helminths infections etc., and suboptimal healthcare infrastructure and access. We have added the below sentence in the discussion (line 348-350) to reiterate this point: 

In addition, there could be some unaccounted factors such as different domestic policies on early childhood development or inequities in healthcare access that could impact the associations between countries.

Reviewer comment: Figure 1: Please add descriptions for different arrow thicknesses and the different box outlines

Response: Thanks, we have now added a description to the figure: “Thicker arrow shows pathway that was not directly assessed as brain development data (dashed box) was not available in the datasets. Other arrows and boxes show the pathways and relationships examined in the analyses”.

---

## [Decision Letter · Decision Letter 1]

2 Feb 2024

Is child anemia associated with early childhood development? A cross-sectional analysis of nine Demographic and Health Surveys

PONE-D-23-01169R1

Dear Dr. Benedict,

We’re pleased to inform you that your manuscript has been judged scientifically suitable for publication and will be formally accepted for publication once it meets all outstanding technical requirements.

Kind regards,

Kannan Navaneetham, PhD

Academic Editor

PLOS ONE

Additional Editor Comments (optional):

Reviewers' comments:

Reviewer's Responses to Questions

**Comments to the Author**

1. If the authors have adequately addressed your comments raised in a previous round of review and you feel that this manuscript is now acceptable for publication, you may indicate that here to bypass the “Comments to the Author” section, enter your conflict of interest statement in the “Confidential to Editor” section, and submit your "Accept" recommendation.

Reviewer #1: All comments have been addressed

2. Is the manuscript technically sound, and do the data support the conclusions?

Reviewer #1: Yes

3. Has the statistical analysis been performed appropriately and rigorously? 

Reviewer #1: Yes

4. Have the authors made all data underlying the findings in their manuscript fully available?

Reviewer #1: Yes

5. Is the manuscript presented in an intelligible fashion and written in standard English?

Reviewer #1: Yes

6. Review Comments to the Author

Reviewer #1: Congratulations on an excellent manuscript! Thank you for so carefully addressing the previous comments. I have no additional comments and look forward to re-reading and sharing with colleagues after it's been officially published.

7. PLOS authors have the option to publish the peer review history of their article (what does this mean?). If published, this will include your full peer review and any attached files.

Reviewer #1: **Yes: **Julie E.H. Nevins

---

## [Editor Report · Acceptance letter]

19 Feb 2024

PONE-D-23-01169R1 

PLOS ONE

Dear Dr. Benedict, 

I'm pleased to inform you that your manuscript has been deemed suitable for publication in PLOS ONE. Congratulations! Your manuscript is now being handed over to our production team.

Kind regards, 

on behalf of

Prof. Kannan Navaneetham 

Academic Editor

PLOS ONE